# DFT, ADMET and Molecular Docking Investigations for the Antimicrobial Activity of 6,6′-Diamino-1,1′,3,3′-tetramethyl-5,5′-(4-chlorobenzylidene)bis[pyrimidine-2,4(1H,3H)-dione]

**DOI:** 10.3390/molecules27030620

**Published:** 2022-01-18

**Authors:** Nesreen T. El-Shamy, Ahmed M. Alkaoud, Rageh K. Hussein, Moez A. Ibrahim, Abdulrahman G. Alhamzani, Mortaga M. Abou-Krisha

**Affiliations:** 1Physics Department, Faculty of Science, Taibah University, Al-Madina Al Munawarah 44256, Saudi Arabia; nesreen_elshamy@yahoo.com or; 2Physics Department, Faculty of Women, Ain Shams University, Cairo 11865, Egypt; 3Physics Department, College of Science, Imam Mohammad Ibn Saud Islamic University (IMSIU), Riyadh 11623, Saudi Arabia; akaoud@imamu.edu.sa (A.M.A.); maimohammed@imamu.edu.sa (M.A.I.); 4Chemistry Department, College of Science, Imam Mohammad Ibn Saud Islamic University (IMSIU), Riyadh 11623, Saudi Arabia; agalhamzani@imamu.edu.sa (A.G.A.); mmaboukrisha@imamu.edu.sa (M.M.A.-K.); 5Department of Chemistry, Faculty of Science, South Valley University, Qena 83523, Egypt

**Keywords:** pyrimidine molecule, DFT, drug likeness, ADMET, molecular docking, antimicrobial activity

## Abstract

Heterocyclic compounds, including pyrimidine derivatives, exhibit a broad variety of biological and pharmacological activities. In this paper, a previously synthesized novel pyrimidine molecule is proposed, and its pharmaceutical properties are investigated. Computational techniques such as the density functional theory, ADMET evaluation, and molecular docking were applied to elucidate the chemical nature, drug likeness and antibacterial function of molecule. The viewpoint of quantum chemical computations revealed that the molecule was relatively stable and has a high electrophilic nature. The contour maps of HOMO-LUMO and molecular electrostatic potential were analyzed to illustrate the charge density distributions that could be associated with the biological activity. Natural bond orbital (NBO) analysis revealed details about the interaction between donor and acceptor within the bond. Drug likeness and ADMET analysis showed that the molecule possesses the agents of safety and the effective combination therapy as pharmaceutical drug. The antimicrobial activity was investigated using molecular docking. The investigated molecule demonstrated a high affinity for binding within the active sites of antibacterial and antimalarial proteins. The high affinity of the antibacterial protein was proved by its low binding energy (−7.97 kcal/mol) and a low inhibition constant value (1.43 µM). The formation of four conventional hydrogen bonds in ligand–protein interactions confirmed the high stability of the resulting complexes. When compared to known standard drugs, the studied molecule displayed a remarkable antimalarial activity, as indicated by higher binding affinity (B.E. −5.86 kcal/mol & Ki = 50.23 M). The pre-selected molecule could be presented as a promising drug candidate for the development of novel antimicrobial agents.

## 1. Introduction

Pyrimidine and its substantial groups of novel derivatives are members of the heterocyclic aromatic family. The pyrimidine molecular structure is primarily composed of a core heteroaromatic ring that included two nitrogen atoms [1,2,3]. Pyrimidines have important pharmacological properties because they are an integral element of DNA and RNA. Additionally, as a crucial element of nucleic acids, it is used as a synthetic forerunner of bioactive molecules [4,5]. Natural product classes with pyrimidine rings are abundant in plants, particularly in vegetables and fruits. These classes contain secondary metabolites with intriguing skeletal characteristics; they have a broad range of biological effects and are extremely important because they are non-toxic [6,7]. In recent years, pyrimidine derivatives have attracted considerable attention because of their variety of pharmacological activities, including anti-epileptic, antiviral, antihistaminic, analgesic, anticonvulsant and anti-inflammatory activities [8,9,10,11,12]. Furthermore, pyrimidine derivatives have numerous antimicrobial applications, such as antibacterial, anti-Toxoplasma, fungicidal and antimalarial applications. Figure 1 depicts the structures of some previously studied drug molecules containing pyrimidine rings [13,14,15,16,17].

Despite advances in antibacterial and antifungal therapeutic interventions, the majority of antimicrobial drugs continue to face multiple challenges. The widespread use of antibiotics has resulted in the emergence of drug-resistant microbes, requiring the quest for novel chemical utilities to address these deficiencies in microbial treatment [18]. The efficacy of such pyrimidine compounds and their derivatives has prompted chemists to search for and synthesize a significant number of active novel drugs to satisfy the demands of the world’s growing health problems. Computational approaches to drug discovery and design have increased in popularity due to its reliability in discovering molecules with high pharmacological properties. The molecular docking method is widely used to determine the proper orientation of drug molecules in the protein-active site as well as to measure their binding affinity. Previously, extensive molecular docking studies were conducted to explore the biological activity of many chemical materials [19,20,21,22]. Drug likeness criteria and an in silico ADMET profile are two other virtual screening tools for the adoption of compounds that may exhibit physiological activity and drug-like identity. The protocols used to estimate drug likeness and ADMET features are produced from a variety of experimental sources of data that have been documented in drug databases [23,24]. There have been numerous theoretical strategies successful in the design and characterization of widely known organic compounds. Density functional theory (DFT) is thought to be the most popular method, with a lower computational cost compared to many other methods. DFT calculations currently yield the most precise and reliable results for different material systems, which are quite compatible with the experimental data [25,26,27,28,29,30]. In the present research, a systematic computational examination of a novel member of pyrimidine compounds was performed by using DFT calculations, ADMET, and a molecular docking procedure. The molecular and optimized structure of the proposed compound is illustrated in Figure 2. The primary objective is to evaluate the antimicrobial activity of presented chemical structure as a potential drug with potent pharmacological properties.

## 2. Results

### 2.1. Frontier Molecular Orbitals

Frontier molecular orbitals (FMOs) are used to predict the most reactive regions in molecular systems and to describe different types of reactions. These FMO properties, such as HOMO and LUMO orbitals, are extremely useful for material science characterization, because they identify the chemical reactivity of the molecules [31]. The HOMO is an electron-rich orbital; it has a tendency to transfer electrons to unoccupied orbitals. LUMO, on the other hand, is an orbital that lacks electrons and can simply accept electrons from other occupied orbitals.

Bearing this in mind, the studied molecule is composed of two building blocks of 6-amino-1,3-dimethylpyrimidine-2,4(1H,3H)-dione connected by a central carbon atom and upper (4-chlorophenyl)methylene group. The HOMO and LUMO molecular orbital distribution is depicted in Figure 3. The HOMO is found primarily on the molecule’s right side, specifically on the majority of the right block (6-amino-1,3-dimethylpyrimidine-2,4(1H,3H)-dione), and then extends slightly to the central carbon atom, followed by the C=C–C bond and the oxygen atom in the molecule’s left block. The LUMO orbitals were located on the left and the upper part of the molecule, i.e., the left block of 6-amino-1,3-dimethylpyrimidine-2,4(1H,3H)-dione and the upper group of (4-chlorophenyl)methylene except the chlorine atom. As a result, HOMO and LUMO analysis revealed a significant susceptibility to charge transfer through the molecule.

### 2.2. Quantum Chemical Descriptors

In the context of quantum chemical calculations, the study of electronic energies has become important in better understanding the nature of molecular chemical stability and reactivity. Koopmans’ theorem represents a theoretical procedure for relating chemical activities of molecular structures to their electronic properties [32]. Quantum chemical descriptors derived from Koopmans’ theorem could be used to identify a molecule’s reactivity, such as ionization potential (I, electronic affinity (A), hardness (η), softness (σ), chemical potential (µ), electronegativity (χ), and electrophilicity (ω). These parameters’ mathematical definitions are as follows:(1)$$\Delta E=E_{\mathrm{LUMO}}-E_{\mathrm{HOMO}}$$
(2)$$I=-E_{\mathrm{HOMO}}\;$$
(3)$$A=-E_{\mathrm{LUMO}}$$
(4)$$\eta =\frac{I-A}{2}$$
(5)$$\sigma =\frac{1}{\eta }=\frac{1}{I-A}$$
(6)$$\chi =\frac{I+A}{2}$$
(7)µ=− χ
(8)$$\omega =\frac{{\;\chi }^{2}}{2\eta }$$

Table 1 shows the HOMO and LUMO energies, as well as the associated chemical descriptors of the studied compound. The HOMO and LUMO energy gap is an indicator of the molecule’s kinetic stability. High chemical stability is attributed to the high energy gap value, whereas high reactivity is associated with a small HOMO-LUMO gap. Ionization potential (I) is the energy required to remove an electron from a molecule’s ground state, and electronic affinity (A) is the energy released when a molecule in its ground state captures an electron. The high value of ionization energy indicates chemical stability, whereas a molecule with a high electronic affinity value is more likely to accept electrons. According to Table 1, the studied compound has good chemical stability due to its large energy gap, high ionization potential, and low electron affinity.

The hardness (η) and softness (σ) are important descriptors for the behavior of a molecule in chemical reaction. Simply, hard molecules have a high resistance to changing their electronic distribution during a reaction, whereas soft molecules have a low resistance to changing their electronic distribution during a reaction. The obtained results revealed a high hardness value plus a low softness value in comparison to similar structures found in the literature [33].

The chemical potential (µ) indicates the possibility of a chemical reaction, the high value of μ (less negative) means it is easier to donate electrons (electron donor), while a small value of μ (more negative) means it is easier to accept electrons (electron acceptor). Electronegativity (χ) of a molecule measures its ability for electron attraction [34]. The chemical potential is calculated to be (−3.29 eV), describing the molecule as donor electrons. Electrophilicity (ω) is a predictor for the electrophilic nature of a chemical species; it measures the propensity of molecule to accept an electron, with high values of ω characterizing good electrophilicity in a molecule. The following is a ranking of organic molecules according to electrophilicity; weak electrophiles have ω less than 0.8 eV, moderate electrophiles have ω in a range between 0.8 and 1.5 eV, and strong electrophiles have ω greater than 1.5 eV [35]. As a result, the calculated electrophilicity value characterizes the investigated compound as a good electrophile. It is common knowledge that drugs with a high electrophilic nature have potent antimicrobial and anticancer properties [36].

### 2.3. Molecular Electrostatic Potential (MEP)

The molecular electrostatic potential (MEP) determines a molecule’s reactivity to nucleophilic and electrophilic attacks by providing information about its nuclear/electronic charge distribution. MEP indicates the molecular contour of electrostatic potential using a color code. Positive, negative, and neutral electrostatic potential are represented by blue, red, and green color, respectively. The blue/red region is the preferred site for nucleophilic/electrophilic attack, or more precisely, the positively/negatively charged sites in the molecule will attract attacking nucleophile/electrophile pieces [37]. Figure 4 illustrates the molecular electrostatic potential of the considered molecule. Positive electrostatic potential (maximum blue color) was found over hydrogen atoms in N-CH3 and N-CH2 groups in both the 6-amino-1,3-dimethylpyrimidine-2,4(1H,3H)-dione blocks, whereas negative electrostatic potential (maximum red color) was scattered over all of the oxygen atoms contained in the molecule. Accordingly, the specified hydrogen atoms are the most reactive sites for nucleophilic attack, while oxygen atoms act as centerpieces for electrophilic attack. The electrostatic potential varies depending on the molecule/drug because of the type and the electronic nature of the constituent atoms. Therefore, the biological activity can be attributed to the difference in the electrostatic potential distribution around the drug [38].

### 2.4. Natural Bond Orbital (NBO) Analysis

In NBO analysis, all possible interactions between “filled” (donor) Lewis-type NBOs and “empty” (acceptor) non-Lewis NBOs were examined, and their corresponding energetic significance was estimated by second-order perturbation theory. As these interactions donate occupancy from the idealized Lewis structure’s localized NBOs into the empty non-Lewis orbitals (and thus, to departures from the idealized Lewis structure description), they are identified as “delocalization” corrections to the zeroth-order natural Lewis structure [39]. The stabilization energy *E*^(2)^ attributed to delocalization (“2e-stabilization”) *i* → *j* is estimated for each donor NBO () and acceptor NBO (*j*) as
$$E^{\left(2\right)}=\Delta E_{ij}=q_{i}\frac{F{\left(i,j\right)}^{2}}{\varepsilon_{j}-\varepsilon_{i}}$$
where *q_i_* is the donor orbital occupancy, *ε_i_*, *ε_j_* are diagonal elements (orbital energies), and *F*(*i*,*j*) is the off-diagonal NBO Fock matrix element. The greater extent of conjugation enjoyed by the molecular system is due to the larger values of the stabilization energy, *E*^(2)^. The delocalization of electron density between a lone pair or bonding and Rydberg or antibonding NBO orbitals increases the stabilizing donor-acceptor interaction.

From the NBO analysis shown in Table 2, the total Lewis structure has 97.74% (core 99.94%, valence Lewis 96.78%) and the total non-Lewis structure has 2.25% (valence non-Lewis 2.04%, Rydberg non-Lewis 0.21%) in this compound. The NBO results showed that the σ (C1-C2) bond was formed by sp^1.56^ hybrid orbital on carbon (60.92% *p*-character) interacting with sp^1.81^ hybrid on carbon (64.37% *p*-character). The sp^1.77^ hybrid on carbon atom (63.87% *p*-character) interacted with sp^1.81^ hybrid of carbon atom (64.41% *p*-character) to form σ (C4-C5). In the same manner, the σ (C12-N18) bond consisted of the sp^2.40^ hybrid on carbon atom (70.56% *p*-character) and sp^1.64^ hybrids of nitrogen (62.08% *p*-character).

The second-order perturbation theory analysis of Fock matrix as a basis for the NBO analysis between the donor and acceptor orbitals of the studied compound are presented in Table 3. The nature of the interaction between the donor (*i*) and acceptor (*j*) is determined by the value of the stabilization energy. The stability of the system increases with increased electron delocalization associated with hyperconjugation and, thus, high stabilization energy. The significant hyperconjugative interactions and the value of their stabilization energies were observed from the NBO analysis. π (C12-C13) → π* (C14-O15) (28.26 Kcal/mol), π (C5-C6) → π* (C2-C3) (19.66 Kcal/mol), σ (C4-C5) → σ* (C1-Cl) (115.48 Kcal/mol), π (C5-C6) → π* (C1-C4) (21.48 Kcal/mol), LP (1 N16) → π* (C21-O30) (65.67 Kcal/mol). These stabilization energies correspond to the stability and the charge transfer within the molecules of the studied compound.

### 2.5. Drug Likeness and ADMET Evaluation

Drug likeness is a calibration for the physicochemical characteristics of a drug with the demanded biopharmaceutical characteristics in human body. According to Lipinski’s five rules, drug likeness is a limitation for consideration of any active drug as a good drug candidate [40,41]. The rules are: molecular weight < 500, (octanol/water coefficient) log P ≤ 5, hydrogen bond donors HBD ≤ 5, hydrogen bond acceptors ≤ 10 [42,43]. ADMET refers to the absorption, distribution, metabolism, excretion, and toxicity of drugs. The optimum ADMET properties are standard descriptors for pharmaceutical characterization in drug selection, and their identification of compatibility for human administration [44].

Table 4 displays the calculated drug likeness and ADMET properties. As shown in the table, the selected ligand followed Lipinski’s five rules, and has the potential to be used as a drug molecule. The studied ligand has a 68% intestinal absorbance, indicating its ease of absorption. Solubility is an important factor in achieving the optimum pharmacological drug. The major issue experienced when synthesizing novel drugs is low solubility. Based on the standard ranking of drug solubility, the calculated water solubility value (log S = −2.51) classified the studied molecule as a good soluble (0 > soluble > −4), which is the desired pharmacokinetics property for absorption and distribution [45]. The absorption results showed that the drug had no effect on P-glycoprotein, being a non-inhibitor of P-glycoprotein I or P-glycoprotein II. The skin permeation Log Kp has a lower value (−8.98), indicating how difficult it is for the skin to absorb the investigated molecule [46].

VDss stands for volume of distribution at steady state, which is the apparent volume of distribution after enough time has passed for the drug to distribute uniformly through all tissues. A high VDss value (>0.5) indicates that the drug is well distributed in the plasma, whereas a low VDss value (<−0.5) indicates that the drug has a poor ability to cross the cell membrane [47]. The predicted VDss value (−0.24) indicated that the drug has reasonable distribution in plasma. Cytochrome P450 is a vital metabolizing enzyme in the human body with five major isoforms: CYP1A2, CYP2C19, CYP2C9, CYP2D6, and CYP3A4. The results showed negative inhibition capability for these enzymes and therefore is safe in pharmacokinetic interactions.

The bioavailability of a drug and its dosing rates to reach to steady-state concentrations is measured by total clearance. The faster the molecule’s excretion process, the higher the total clearance value. Organic cation transporter 2 (OCT 2) is a protein transporter that plays an important role in the renal drug clearance. The inhibition of OCT 2 substrates by drug molecules may result in adverse reactions. The studied molecule has a low total clearance value and non-inhibitor for of OCT 2. AMES toxicity is used to determine whether a molecule is mutagenic or not. Inhibiting potassium channels via hERG inhibitors could lead to catastrophic diseases. The results showed that the mentioned molecule is not mutagenic and has non-inhibitory property of hERG I/II.

Bioavailability Radar is used to provide a quick assessment of drug likeness. The Radar 2D image is divided into six partitions, each representing a different physicochemical property, such as lipophilicity, size, polarity, solubility, flexibility, and saturation. The red line of the examined compound must be entirely contained within the pink region to be classified as drug-like, disadvantageous physicochemical property can be identified in the deviation outside the pink region [48]. As shown in Figure 5, the Radar plot displays a zone with the optimal range of drug likeness, with the exception of a minor polarity deviation.

### 2.6. Molecular Docking

The docking results for assessing the antibacterial and antimalarial activity are shown in Table 5. The calculated binding affinities confirmed the strong ability of the target protein to form stable complexes. This is proved by lower binding energies (B.E.) being obtained compared to similar structures found in studies in the literature [49]. The inhibition constant (Ki) is a measure of the therapeutic potential of a drug to inhibit the activity of an enzyme, with higher binding affinity being associated with lower values of Ki. The docking results show a strong antibacterial activity, with a binding energy value of −7.97 kcal/mol and a minimum inhibition constant of 1.43 µM. The docking results of the DHFR protein (1U72), a well-known target for antibacterial drugs, demonstrated a higher binding affinity than previous studies evaluating the antimicrobial activity of pyrimidine derivatives. Among the many compounds assessed in this study, the best docking score was −6.83 kcal/mol [50], while our investigated molecule achieved 7.00 kcal/mol.

H-bonding is a notable indicator of strong protein–ligand interactions, and it commonly results in high binding affinity [51]. In protein–ligand interactions, the number of hydrogen bonds often increases the inhibitor potency against the target protein. Figure 6 presents a 2D representation of the binding modes within the target protein’s active site. The formation of four conventional hydrogen bonds with antibacterial and antimalarial target proteins resulted in good ligand binding. In the docked molecule, hydrogen bonds were primarily formed with =O and NH2 groups. Remarkably, in Figure 6A,D, one of the two building blocks is twisted, resulting in a molecule with a symmetric form on both sides. This is due to the same residue (TYR 41 in Figure 6A and ILE 357 in Figure 6D) forming two hydrogen bonds with the H-atoms of the two NH2 groups, which are far apart.

Furthermore, as a standard comparison with the studied molecule, the antibacterial and antimalarial drug (Trimethoprim), which contains a pyrimidine ring, was docked into the same investigated protein receptors. Trimethoprim outperforms the ligand in bacterial activity, with the exception of inhibiting the 3ACX protein, where the ligand recorded a binding energy (−7.97 kcal/mol) that was close to that of Trimethoprim (−8.09 kcal/mol). Remarkably, the studied compound inhibited the malarial target more effectively than Trimethoprim. The ligand had a higher docking score, a more stable binding energy (−5.86 kcal/mol), and a much lower inhibition constant (50.23 M) than Trimethoprim.

## 3. Materials and Methods

### 3.1. Experimental Details

The molecule 6,6′-Diamino-1,1′,3,3′-tetramethyl-5,5′-(4-chlorobenzylidene)bis[pyrimidine-2,4(1H,3H)-dione], which has the chemical formula C19H21ClN6O4, was previously synthesized by Das et al. due to the biological activity and medicinal applications of heterocyclic compounds, particularly pyrimidine derivatives. To fully dissolve 6-Amino-1,3-dimethylpyrimidine-2,4(1H,3H)-dione (2 mmol), distilled water was added in excess. After this, p-chlorobenzaldehyde was added until a precipitate formed. The mixture was filtered and purified by column chromatography after 1 h of stirring. Using distilled ethanol, the product was recrystallized, obtaining appropriate crystals for data collection [52]. The crystallographic data for this compound can be found in the Appendix A.

Staphylococcus aureus bacteria (*S. aureus*) contain a diverse set of enzymes that are considered necessary for *S. aureus* growth and reproduction. One of these enzymes that has been recognized as an antibacterial target is carotenoid dehydrosqualene synthase (CrtM) [53]. PBP2a is an enzyme that contributes ranspeptidase activity to the biosynthesis of Methicillin-resistant Staphylococcus aureus (MRSA) cell walls, and it represents a promising target for new antibiotics that inhibit bacterial cell wall biosynthesis [54]. The enzyme dihydrofolate reductase (DHFR) has been identified as a therapeutic target for the treatment of diverse microbial, tumors, protozoal and cancer diseases. DHFR is required for folate metabolism as well as purine and thymidylate synthesis during cell proliferation. Antibacterial DHFR inhibitors work by inhibiting the synthesis of RNA and proteins, resulting in cellular damage [55,56]. Plasmodium falciparum dihydrofolate reductase-thymidylate synthase (pfDHFR-TS) is a well-known homodimeric enzyme with DHFR and thymidylate synthase (TS) domain residues. pfDHFR-TS produces folates and thymidylate (dTMP), both of which are required for DNA synthesis, and has been studied extensively in order to develop effective antimalarial drugs [57].

CrtM (PDB code 3ACX), PBP2a (PDB code 1VQQ) and DHFR (PDB code 1U72) were selected as antibacterial target proteins, while pfDHFR-TS (PDB code 3QGT) was set for antimalarial investigation. The crystal structures of the mentioned proteins were obtained in.pdb format from the Protein Data Bank archive.

### 3.2. Computational Methods

The DFT method, represented in the model of Becke-3 Parameter-Lee-Yang-Parr (B3LYP) and 6-311** basis set [58,59], was used to optimize the title molecule using the Gaussian09 program package [60]. The output data, such as the optimized geometry, HOMO-LUMO and MEP, were visualized using the Gauss View 6 graphical interface [61]. Natural bond orbitals (NBOs) were calculated by taking into account all possible interactions between filled donor and empty acceptor natural orbits, and their energetic importance was predicted using second-order perturbation theory. The NBO program included in the Gaussian 09 package was used for NBO calculations [62]. Many popular websites are now widely used to provide pharmacokinetics information. In this investigation, two online servers, pkCSM and SwissADME, were used to evaluate the ADMET and drug likeness properties [48,63].

For molecular docking computation, Auto Dock Tools 1.5.6 [64] was used to predict the binding affinities and type of interactions between the ligand and target proteins. The steps of protein receptor preparation started with the removal of water molecules, the addition of hydrogen atoms and the assignment of Gasteiger charges to receptor atoms. The protein and ligand rigid pdbqt files were created, and grid box was positioned to encompass the outlined active pocket on the receptor surface. The docking calculation was preceded using AutoDock Tool, with the Lamarckian genetic algorithm (LGA) to find the best conformation pose of ligand within the active site of the protein. The ligand–protein interactions were extracted and visualized using the Discovery Studio 4.5 software [65].

## 4. Conclusions

A comprehensive study was carried out to determine the chemical stability and antimicrobial activity of 6,6′-Diamino-1,1′,3,3′-tetramethyl-5,5′-(4-chlorobenzylidene)bis[pyrimidine-2,4(1H,3H)-dione]. Using the DFT approach, the quantum chemical descriptors recognized the title molecule as stable structure, and the calculated electrophilic index was 2.17 eV. MEP distribution revealed the potential sites for electrophilic and nucleophilic attacks. The compound satisfied the required criteria of drug likeness and ADMET profile for being a drug with good pharmaceutical properties. According to the results of the molecular docking study, the title molecule is biologically active against bacterial proteins due to its lower binding free energy. The most effective antibacterial activity was against the carotenoid dehydrosqualene synthase (CrtM) enzyme, with B.E. −7.97 kcal/mol and Ki 1.43 µM. The studied molecule exhibited remarkable antimalarial activity, as evidenced by higher binding affinity (B.E. −5.86 kcal/mol & Ki = 50.23 µM) than standard antimicrobial drugs. The configuration of conventional hydrogen bonds in the active pocket of the target protein confirmed the molecule’s strong inhibitory property. These findings indicate that the current molecule has promising action as an antimicrobial agent.

## Figures and Tables

**Figure 1 molecules-27-00620-f001:**
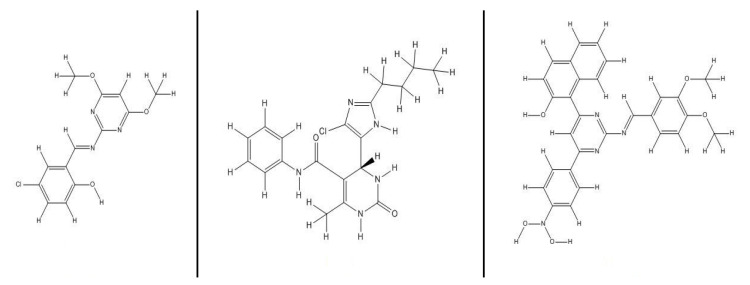
Drug molecules have a pyrimidine ring in their molecular structure.

**Figure 2 molecules-27-00620-f002:**
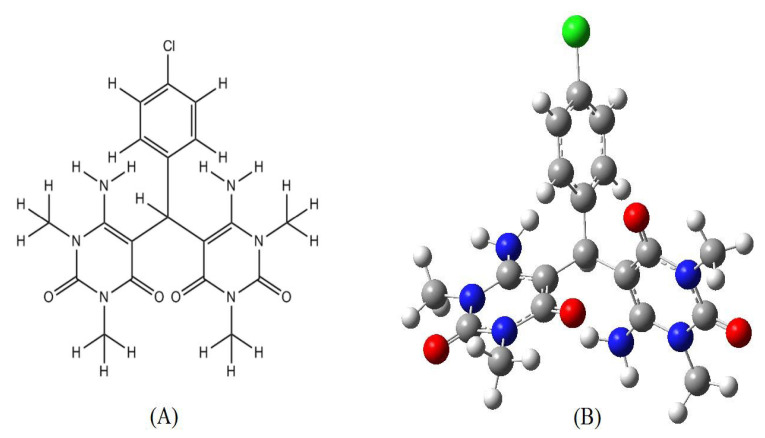
The molecular structure (**A**) and the optimized structure (**B**) of the proposed compound.

**Figure 3 molecules-27-00620-f003:**
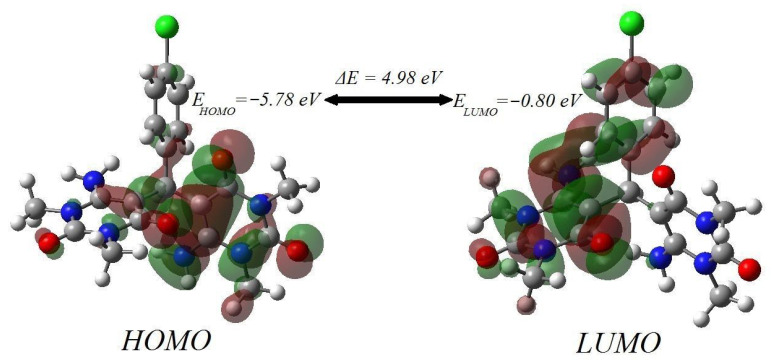
The pattern of HOMO and LUMO frontier molecular orbital surfaces.

**Figure 4 molecules-27-00620-f004:**
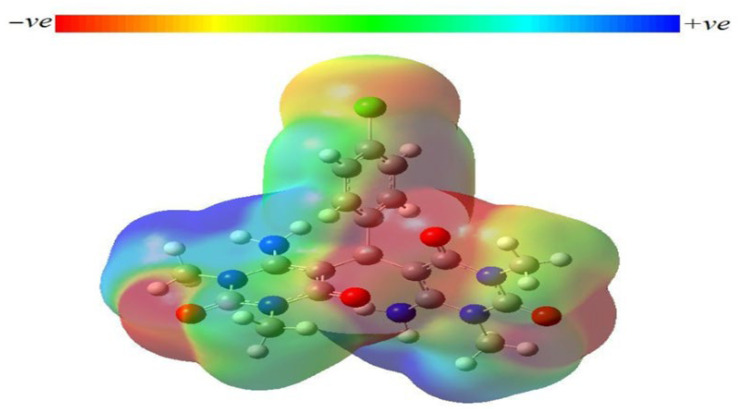
Molecular electrostatic potential (MEP) 3D plot of the investigated molecule.

**Figure 5 molecules-27-00620-f005:**
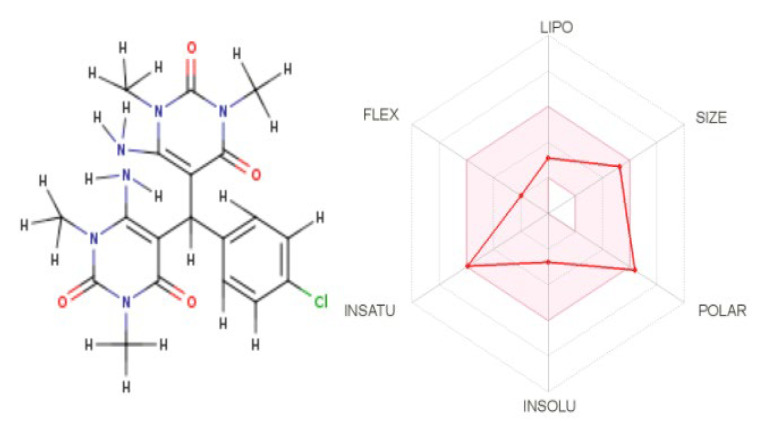
Bioavailability radar depiction of the two-dimensional molecular structure.

**Figure 6 molecules-27-00620-f006:**
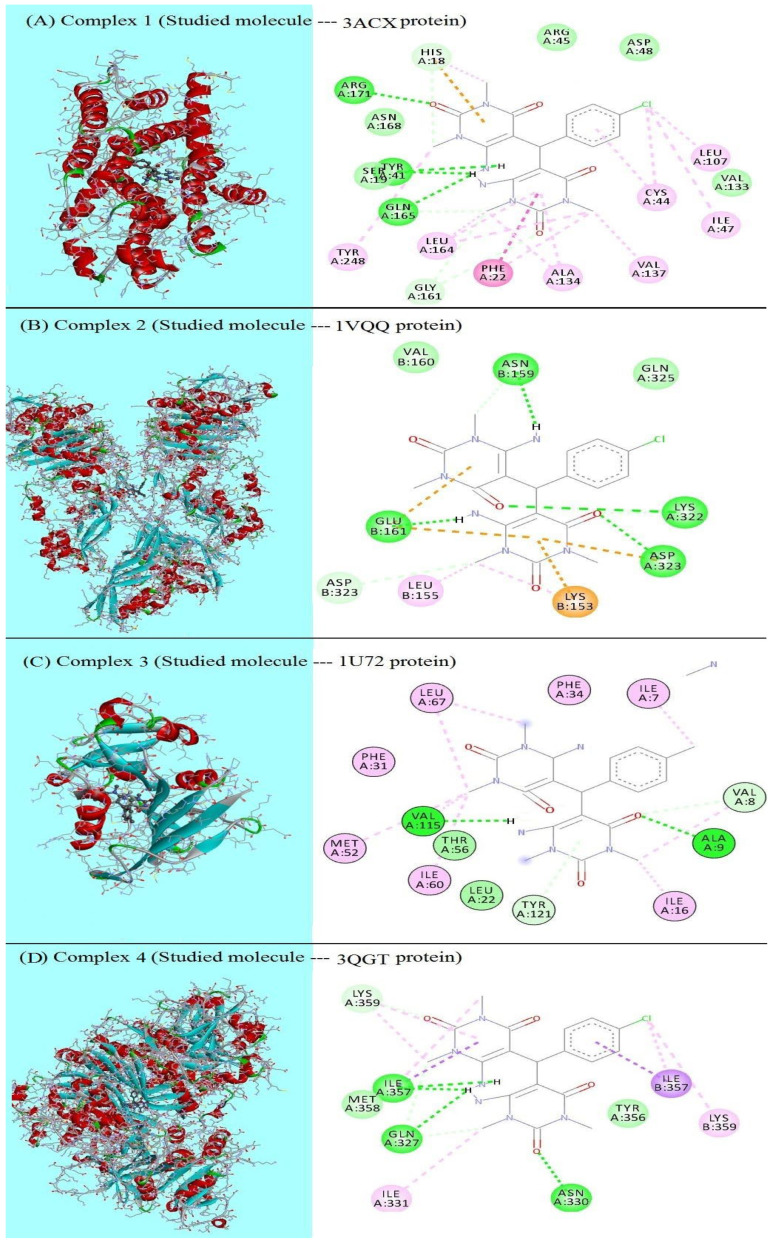
3D, 2D representations of the interactions of the studied molecule inside the active sites of the target proteins.

**Table 1 molecules-27-00620-t001:** The calculated HOMO, LUMO energies and their corresponding quantum chemical parameters.

Parameter	Symbol	Value	Unit
HOMO Energy	E_HOMO_	−5.78	(eV)
LUMO Energy	E_LUMO_	−0.80	(eV)
Energy gap	ΔE	4.98	(eV)
Ionization potential	I	5.78	(eV)
Electron affinity	A	0.80	(eV)
Chemical hardness	η	2.49	(eV)
Chemical softness	σ	0.40	(eV)^−1^
Electronegativity	χ	3.29	(eV)
Chemical potential	µ	−3.29	(eV)
Electrophilicity	ω	2.17	(eV)

**Table 2 molecules-27-00620-t002:** Occupancy of natural orbitals and hybrids of compound for C,N,O,H,Cl atoms.

Parameters	Occupancies (e)	Hybrids	Atomic Orbitals %
σ C1-C2	1.97901	sp^1.56^	s (39.04%) p (60.92%) d (0.04%)
σ C1-C4	1.97896	sp^1.56^	s (39.08%) p (60.88%) d (0.04%)
σ C1-Cl1	1.98611	sp^3.60^	s (21.72%) p (78.10%) d (0.19%)
σ C2-C3	1.96743	sp^1.75^	s (36.39%) p (63.56%) d (0.04%)
σ C2-H8	1.97753	sp^2.57^	s (28.00%) p (71.96%) d (0.05%)
σ C3-C6	1.97158	sp^1.72^	s (36.74%) p (63.22%) d (0.04%)
σ C3-H9	1.97563	sp^2.61^	s (27.69%) p (72.26%) d (0.05%)
σ C4-C5	1.96730	sp^1.77^	s (36.09%) p (63.87%) d (0.04%)
σ C4-H10	1.97771	sp^2.56^	s (28.08%) p (71.87%) d (0.05%)
σ C5-H7	1.97738	sp^2.68^	s (27.18%) p (72.77%) d (0.05%)
σ C6-C31	1.96267	sp^2.20^	s (31.28%) p (68.69%) d (0.02%)
σ C12-N18	1.98760	sp^2.40^	s (29.35%) p (70.56%) d (0.09%)
σ C14-O15	1.99226	sp^2.07^	s (32.55%) p (67.31%) d (0.13%)
σ C26-H28	1.98897	sp^2.98^	s (25.13%) p (74.80%) d (0.07%)
LP (1) O15	1.97587	sp^0.72^	s (58.12%) p (41.86%) d (0.02%)
LP (1) O51	1.97556	sp^0.72^	s (58.01%) p (41.97%) d (0.02%)
σ* C1-Cl1	0.03496	sp^3.60^	s (21.72%) p (78.10%) d (0.19%)
σ* C4-H10	0.01433	sp^2.56^	s (28.08%) p (71.87%) d (0.05%)
σ* C31-C32	0.03013	sp^2.59^	s (27.83%) p (72.13%) d (0.04%)
σ* N39-C41	0.07580	sp^1.93^	s (34.09%) p (65.84%) d (0.06%)
σ* C47-H50	0.01098	sp^2.87^	s (25.85%) p (74.09%) d (0.06%)

**Table 3 molecules-27-00620-t003:** Second-order perturbation theory analysis of Fock matrix as a basis for the NBO analysis between donor and acceptor orbitals of the studied compound.

Type	Donor(i)	Type	Acceptor(j)	*E*^(*2*)^ Kcal/mol	*E*(*i*)−*E*(*j*) (a.u)	*F*(*i*,*j*) (a.u)
π	C12-C13	π*	C14-O15	28.26	0.3	0.086
π	C5-C6	π*	C1-C4	21.48	0.27	0.069
π	C5-C6	π*	C2-C3	19.66	0.29	0.068
π	C38-O51	π*	C32-C34	4.54	0.38	0.04
π	C12-C13	σ*	C6-C31	3.52	0.69	0.046
π	C12-C13	σ*	C31-H33	2.01	0.67	0.034
π	C32-C34	σ*	C31-H33	1.21	0.66	0.026
π	C38-O51	π*	C38-O51	1.08	0.38	0.02
π	C32-C34	σ*	N35-H36	0.53	0.67	0.018
σ	C4-C5	σ*	C1-Cl11	5.48	0.84	0.061
σ	C3-H9	σ*	C5-C6	4.82	1.08	0.064
σ	C31-C32	σ*	C34-N40	4.77	1.04	0.063
σ	C32-C38	σ*	C32-C34	4.29	1.27	0.066
σ	C1-C2	σ*	C1-C4	4.12	1.29	0.065
σ	N18-H20	σ*	C12-C13	4.02	1.26	0.064
σ	C13-C14	σ*	N16-C26	3.16	1.01	0.051
σ	C12-C13	σ*	C12-N18	3.11	1.16	0.054
σ	N40-C41	σ*	N39-C43	2.61	1.14	0.049
σ	N17-C21	σ*	C12-N17	2.19	1.24	0.047
σ	C38-N39	σ*	N39-C43	1.16	1.11	0.032
σ	C5-H7	σ*	C4-C5	0.64	1.09	0.024
σ	C38-N39	σ*	C43-H44	0.51	1.21	0.022
LP(1)	N16	π*	C21-O30	65.67	0.26	0.117
LP(1)	N40	π*	C32-C34	45.98	0.3	0.106
LP(2)	O15	σ*	C14-N16	28.83	0.62	0.121
LP(3)	Cl11	π*	C1-C4	12.11	0.33	0.062
LP(1)	N40	σ*	C47-H49	5.59	0.64	0.059
LP(1)	O51	σ*	C32-C38	2.88	1.16	0.052
LP(1)	O30	σ*	N17-C21	1.98	1.07	0.042
LP(2)	O51	σ*	C43-H44	0.91	0.67	0.023
LP(2)	O51	π*	C2-C3	0.62	0.27	0.012

LP—Lone pair, *E*^(*2*)^—Stabilization energy, *E*(*i*)*−E*(*j*)—Energy difference between the donor and acceptor NBO orbitals, *F*(*i,j*)—Fock matrix element between i and j NBO orbitals.

**Table 4 molecules-27-00620-t004:** The drug likeness profile and ADMET analysis.

Property	Parameter/Model Name	Predicted Value	Unit
Drug likeness	MW	432.86	(g/mol)
Drug likeness	HBD	2	Numeric
Drug likeness	HBA	10	Numeric
Drug likeness	nRotb	3	Numeric
Drug likeness	octanol/water coefficient	2.69	Numeric (Log Po/w)
Absorption	Water solubility	−2.51	Numeric (log mol/L)
Absorption	Intestinal absorption	68.56	Numeric (% Absorbed)
Absorption	P-glycoprotein I inhibitor	No	Categorical (Yes/No)
Absorption	P-glycoprotein II inhibitor	No	Categorical (Yes/No)
Absorption	Skin Permeability	−8.98	Numeric (log Kp)
Distribution	VDss (human)	−0.239	Numeric (log L/kg)
Metabolism	CYP1A2 inhibitor	No	Categorical (Yes/No)
Metabolism	CYP2C19 inhibitor	No	Categorical (Yes/No)
Metabolism	CYP2C9 inhibitor	No	Categorical (Yes/No)
Metabolism	CYP2D6 inhibitor	No	Categorical (Yes/No)
Metabolism	CYP3A4 inhibitor	No	Categorical (Yes/No)
Excretion	Total Clearance	−0.62	Numeric (log ml/min/kg)
Excretion	Renal OCT2 substrate	No	Categorical (Yes/No)
Toxicity	AMES toxicity	No	Categorical (Yes/No)
Toxicity	Max. tolerated dose (human)	−0.004	Numeric (log mg/kg/day)
Toxicity	hERG I inhibitor	No	Categorical (Yes/No)
Toxicity	hERG II inhibitor	No	Categorical (Yes/No)
Toxicity	Oral Rat Acute Toxicity (LD50)	2.746	Numeric (mol/kg)
Toxicity	Oral Rat Chronic Toxicity (LOAEL)	0.932	Numeric (log mg/kg_bw/day)
Toxicity	Hepatotoxicity	Yes	Categorical (Yes/No)
Toxicity	Skin Sensitisation	No	Categorical (Yes/No)
Toxicity	Pyriformis toxicity	0.294	Numeric (log ug/L)
Toxicity	Minnow toxicity	3.052	Numeric (log mM)

**Table 5 molecules-27-00620-t005:** The achieved docking results for the complexes of the studied molecule (C_19_H_21_ClN_6_O_4_) and standard drug (Trimethoprim) with antibacterial, antimalarial target proteins.

Antimicrobial Activity	Target Protein	Docked Molecule	Binding Energy (kcal/mol)	Inhibition Constant (µM)	Hydrogen Bonds	Interacting Residues	Bond Distance (Å)
**Antibacterial**	3ACX	C_19_H_21_ClN_6_O_4_	−7.97	1.43	4	ARG171	6.43
						TYR41	5.21
						SER19	5.99
						GLN165	3.54
		Trimethoprim	−8.09	1.18	3	VAL133	3.86
						ASP48	3.57
						CYC44	4.06
**Antibacterial**	1VQQ	C_19_H_21_ClN_6_O_4_	−6.43	19.23	4	ASN159	3.66
						GLU161	3.82
						ASP323	3.74
						LYS322	3.60
		Trimethoprim	−7.66	2.43	4	ASP323	4.30
						ASP323	5.00
						GLN 325	4.40
						GLU161	5.16
**Antibacterial**	1U72	C_19_H_21_ClN_6_O_4_	−7.00	7.4	2	VAL115	3.71
						ALA9	3.80
		Trimethoprim	−7.78	1.97	3	GLU30	3.95
						VAL8	5.23
						TYR121	6.05
**Antimalarial**	3QGT	C_19_H_21_ClN_6_O_4_	−5.86	50.23	4	ASN330	3.89
						GLN327	4.24
						ILE357	4.62
						ILE357	4.63
		Trimethoprim	−5.19	156.88	5	GLN327	3.65
						GLN327	3.95
						LYS359	4.85
						LYS359	5.09
						ILE357	4.31

## Data Availability

The following data are available online; The vibrational frequency results obtained from the optimized structural parameters by using the same level of calculations (B3LYP method with 6-311G** basis set), as well as the crystallographic data of the studied compound.

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
