# Peer review of "DFT, ADMET and Molecular Docking Investigations for the Antimicrobial Activity of 6,6′-Diamino-1,1′,3,3′-tetramethyl-5,5′-(4-chlorobenzylidene)bis[pyrimidine-2,4(1H,3H)-dione]"

_molecules, 2022, doi:10.3390/molecules27030620_

Round 1

Reviewer 1 Report

      The authors utilized the computational tools for assessing the DFT, ADMET and Molecular Docking behaviour of previously synthesized compound 6,6'-Diamino-1,1', 3,3'-tetramethyl-5,5'-(4-chlorobenzylidene) bis [pyrimidine-2, 4(1H,3H)-dione] by Das et al., to consider the target molecule as a prospective antimicrobial lead candidate. The following needs to be revised for possible publication.

  1. Need to redraw the structure of the target compound using ChemDraw. It seems the structure is copied from reference-44.
  2. Incorporate a figure containing the structures of some drug molecules with pyrimidine ring in the introduction part. Additionally, add a line about the figure.
  3. Incorporate the synthetic scheme how the target compound was synthesized by looking into reference-44 by Das et al.,
  4. Methods needs to be elaborated. It is too short.
  5. Why the authors have docked the compound on the antimalarial protein? Justify.
  6. What is the rationale for selecting the proteins 3ACX and 1VQQ for assessing the antimicrobial activity by molecular docking? Justify with appropriate references.
  7. Drug molecules containing pyrimidine scaffold better interact with dihydrofolate reductase (DHFR) enzyme. I strongly suggest the authors to dock the compound against bacterial dihydrofolate reductase enzyme and compare its activity with standard antimicrobial drugs containing pyrimidine ring that act through DHFR inhibition.
  8. Include inhibition constant (Ki) and docking score in conclusions.
  9. Minor grammatical errors need to be corrected.
  10. Some of the references lack doi’s.

Author Response

Dear reviewer

You find  a point-by-point response to your comments in the uploaded file.

                                       Sincerely Regards

Reviewer 2 Report

DFT, ADMET and Molecular Docking Investigations for the Antimicrobial Activity of 6,6'-Diamino-1,1', 3,3'-tetramethyl- 35,5'-(4-chlorobenzylidene) bis [pyrimidine-2, 4(1H,3H)-dione]

  1. Although, the compound was NOT synthesized by the authors, the experimental spectroscopic information such FT-IR should be compared with theory FT-IR through the potential energy distribution analysis
  2. In like manner, the schematic synthetic route should be include in the experimental detail section
  3. The quantum chemical parameters define in Table 1 should be removed and define outside
  4. Docking validation should be conducted using the Ramachandra plot
  5. Molecular docking using standard antibacterial drug should be conducted and compared with the studied compound
  6. Fukui function for the prediction of reactivity should be included
  7. Natural bond orbital analysis for the investigation of intramolecular charge transfer should be conducted
  8. The discussion should be improve by citing the following articles:
    • Obu, Q. S., Louis, H., Odey, J. O., Eko, I. J., Abdullahi, S., Ntui, T. N., & Offiong, O. E. (2021). Synthesis, Spectra (FT-IR, NMR) investigations, DFT study, in silico ADMET and Molecular docking analysis of 2-amino-4-(4-aminophenyl) thiophene-3-carbonitrile as a potential anti-tubercular agent. Journal of Molecular Structure, 130880.
    • Agwupuye, J. A., Neji, P. A., Louis, H., Odey, J. O., Unimuke, T. O., Bisiong, E. A., ... & Ntui, T. N. (2021). Investigation on electronic structure, vibrational spectra, NBO analysis, and molecular docking studies of aflatoxins and selected emerging mycotoxins against wild-type androgen receptor. Heliyon7(7), e07544.
    • Erol, M., Celik, I., & Kuyucuklu, G. (2021). Synthesis, Molecular Docking, Molecular Dynamics, DFT and Antimicrobial Activity Studies of 5-substituted-2-(p-methylphenyl) benzoxazole Derivatives. Journal of Molecular Structure1234, 130151.
    • Khan, I. M., Islam, M., Shakya, S., Alam, N., Imtiaz, S., & Islam, M. R. (2021). Synthesis, spectroscopic characterization, antimicrobial activity, molecular docking and DFT studies of proton transfer (H-bonded) complex of 8-aminoquinoline (donor) with chloranilic acid (acceptor). Journal of Biomolecular Structure and Dynamics, 1-15.
    • Mary, Y. S., Mary, Y. S., Krátký, M., Vinsova, J., Baraldi, C., & Gamberini, M. C. (2021). DFT, molecular docking and SERS (concentration and solvent dependant) investigations of a methylisoxazole derivative with potential antimicrobial activity. Journal of Molecular Structure1232, 130034.

Author Response

(The authors gave the same response as above.)

Reviewer 3 Report

The authors have investigated the DFT, ADMET and Molecular Docking of

of 6,6'-Diamino-1,1', 3,3'-tetramethyl- 35,5'-(4-chlorobenzylidene) bis [pyrimidine-2, 4(1H,3H)-dione]. Although the presented in detail bonding interaction and ADMET properties are very interesting. I would like to recommend acceptance after minor revision.

(1).

Authors should refer to the articles that synthesized the compound described DFT, ADMET, and Docking properties. And what is the rationalization to select this compound for analysis?

(2).

Should the authors explain the solubility of the reported compound? And how it will be affected for the Pharmacokinetics (PK) studies.

(3)

Authors should also refer to the biological importance of natural product-based pyrimidine compounds (for example, see DOI: 10.1007/s11030-015-9621-3 &

doi:10.1007/s00706-017-2024-7.

(4).

 Co-crystallographic data will give more information about the potential binding site interactions. Have you been investigated the study? 

Author Response

(The authors gave the same response as above.)

Round 2

Reviewer 1 Report

The manuscript is much improved and the comments were addressed appropriately. It is now suitable for publication.

Reviewer 2 Report

Manuscript now Okay